# Learning Deep Attribution Priors Based On Prior Knowledge

**Ethan Weinberger**
Paul G. Allen School of Computer Science
University of Washington
Seattle, WA 98195
ewein@cs.washington.edu

**Joseph D. Janizek**
Paul G. Allen School of Computer Science
University of Washington
Seattle, WA 98195
jjanizek@cs.washington.edu

**Su-In Lee**
Paul G. Allen School of Computer Science
University of Washington
Seattle, WA 98195
suinlee@cs.washington.edu

## Abstract

Feature attribution methods, which explain an individual prediction made by a model as a sum of *attributions* for each input feature, are an essential tool for understanding the behavior of complex deep learning models. However, ensuring that models produce meaningful explanations, rather than ones that rely on noise, is not straightforward. Exacerbating this problem is the fact that attribution methods do not provide insight as to *why* features are assigned their attribution values, leading to explanations that are difficult to interpret. In real-world problems we often have sets of additional information for each feature that are predictive of that feature's importance to the task at hand. Here, we propose the *deep attribution prior* (DAPr) framework to exploit such information to overcome the limitations of attribution methods. Our framework jointly learns a relationship between prior information and feature importance, as well as biases models to have explanations that rely on features predicted to be important. We find that our framework both results in networks that generalize better to out of sample data and admits new methods for interpreting model behavior.

## 1 Introduction

Recent advances in machine learning have come in the form of complex models that humans struggle to interpret. In response to the black-box nature of these models, a variety of recent work has focused on model interpretability [11]. One particular line of work that has gained much attention is that of feature attribution methods [22, 37, 28, 3, 33]. Given a model and a specific prediction made by that model, these methods assign a numeric value to each input feature, indicating how important that feature was for the given prediction (Figure 1a). A variety of such methods have been proposed, and previous work has focused on how such methods can be used to gain insight into model behavior in applications where model trust is critical [43, 31]

Given a set of attributions, a natural question is *why* a feature was assigned a specific attribution value. In some settings it is easy for a human to evaluate the sensibility of attributions; for example, in image classification problems we can overlay attribution values on the original image. However, in many other domains we do not have the ability to assess the validity of attribution values so

easily. Recent work [30, 7, 29] has attempted to address this problem by regularizing model training so that both model predictions and explanations agree with prior human knowledge. While such regularization methods do lead to noticeably different attributions, they suffer from two shortcomings. First, they require a human expert with prior knowledge about a given problem domain to construct a domain-specific regularization function. Second, after a set of feature attribution values is produced for a given prediction, we still do not have a human-interpretable way to understand why that set of values was produced beyond trusting in the regularization procedure.

At the same time, the overparameterization of deep learning models makes them prone to overfitting to noise. This limitation has stalled the adoption of deep models in domains where data acquisition is difficult, such as many areas of medicine [5]. As such, methods that encourage deep models to learn more generalizable representations, even when sample sizes are small, are in high demand.

In many real-world tasks we have access to sets of information about each feature that characterize the feature. We refer to such sets of information as *meta-features*. Meta-features can potentially encode information on a feature's relative importance across all samples to the task at hand. For example, we can consider a task where we seek to predict a user's rating for a new movie based on their previous ratings for other movies. In this case we would expect ratings for movies with similar meta-feature values to be more relevant than those for dissimilar films. Potential meta-features to capture the similarity between features in this task could include movie genre and director, among others. By using meta-features to bias models to focus on more relevant features when making predictions, we may be able to achieve both greater model accuracy and interpretability. For a given problem, while a practitioner may have an idea of potential meta-features that are relevant to the problem, it is far less likely that they know *a priori* a specific relationship between these meta-features and global feature importance. As a first step towards mitigating this problem, we propose the *deep attribution prior* (DAPr) framework for training deep neural networks that simultaneously learns a relationship between meta-features and global feature importance values, and biases the prediction model to have local explanations that broadly agree with this learned relationship (Figure 1b). For clarity, in the remainder of the text we use feature importance specifically to refer to a feature's global relevance to a task across samples, and attribution value to refer to its local relevance for a specific prediction.

We apply our method to a synthetic dataset and two real-world biological datasets. Empirically we find that models trained using the DAPr framework achieve better performance on tasks where training data is limited. Furthermore, we demonstrate that such learned meta-feature to feature importance relationships can be leveraged to obtain more insight into the behavior of deep models.

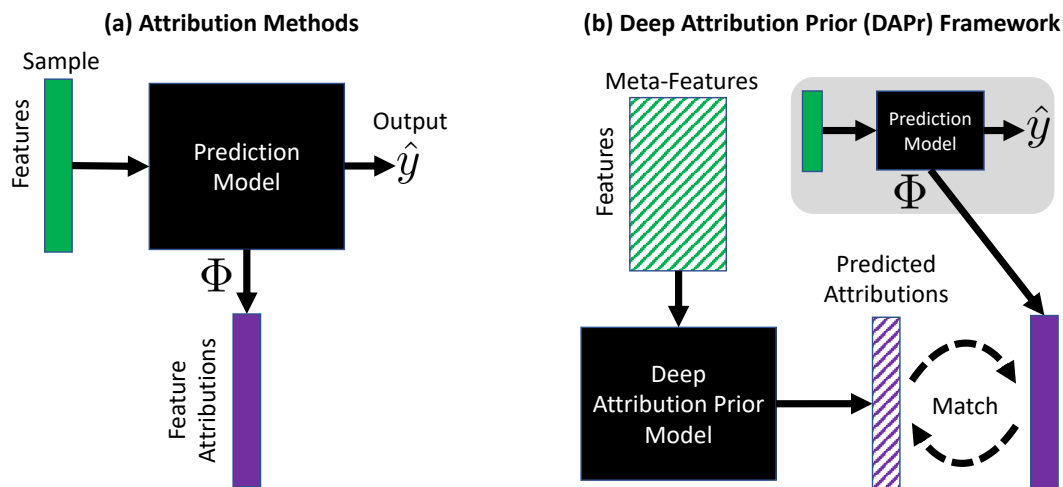

Figure 1: (a) An attribution method $\Phi$ is used to explain the decision of an arbitrary black-box model. (b) Overview of the DAPr framework. In addition to training a prediction model, we train a second prior model to use meta-features to predict global feature importance values. We also add a penalty term to our prediction model's loss function that encourages the values of local attributions to be similar to predicted global importance values.

## 2 Related Work

Prior knowledge has long been used to guide the design of new machine learning algorithms. In a deep learning context, domain-specific knowledge has been essential to the design of model architectures that have achieved breakthroughs in performance. For example, LeCun et al. [17] took advantage of the smoothness of images to design a specialized neural network architecture for vision problems. Similarly, Bahdanau et al. [4] introduced the attention mechanism to alleviate problems with memorizing long source sentences in neural machine translation.

A line of recent work has studied using prior knowledge to bias the training of previously-existing network architectures so that their *explanations* align with prior knowledge. Such work [7, 30, 29] has focused on training networks by penalizing differences between local feature importance, as measured by an attribution method, and a function constructed by a human domain expert.

One particular form of prior knowledge is that of meta-features. Previous work has studied using meta-features to constrain learning algorithms to learn more generalizable models. Lee et al. [18] devised an algorithm for linear models that uses meta-features to transfer knowledge across related tasks, assuming that a feature's weight is a function of its meta-feature values. Krupka and Tishby [15] proposed a framework for training support vector machines using meta-features to define a prior on the prediction model's weights. More recently Lee et al. [19] proposed the MERGE algorithm, which trains a linear model so that its weights match the output of a second linear model that learns a relationship between feature weights and meta-feature values. The two models are iteratively optimized until convergence. However, to our knowledge DAPr is the first method to bias the training of neural networks using prior knowledge in the form of meta-features.

## 3 Deep Attribution Priors

In this section we introduce a formal definition of a deep attribution prior by extending that of the attribution prior proposed in Erion et al. [7]. Let $X \in \mathbb{R}^{n \times p}$ denote a training dataset consisting of $n$ samples, each of which has $p$ features. Similarly, let $Y \in \mathbb{R}^n$ denote labels for the samples in our training dataset. In a standard model training procedure, we wish to find the model $\hat{f}$ in some model class $\mathcal{F}$ that minimizes the average prediction loss on our dataset, as determined by some loss function $\mathcal{L}$. In order to avoid overfitting on training data, a regularization function $\Omega'$ on the prediction model's parameters $\theta_f$ is often included, giving the equation

$$\hat{f} = \underset{f \in \mathcal{F}}{\operatorname{argmin}} \frac{1}{n} \sum_{x,y} \mathcal{L}(f(x), y) + \lambda' \Omega'(\theta_f),$$

where $\lambda'$ is a scalar hyperparameter controlling the strength of the regularization. For a given feature attribution method $\Phi(\theta, x)$, an attribution prior is defined as some function $\Omega \colon \mathbb{R}^p \to \mathbb{R}$ that assigns scalar-valued penalties to the feature attributions for $f$ with input $x$. This leads us to the following objective:

$$\hat{f} = \underset{f \in \mathcal{F}}{\operatorname{argmin}} \frac{1}{n} \sum_{x,y} \mathcal{L}(f(x), y) + \lambda \sum_{x} \Omega(\Phi(\theta_f, x)).$$

Now suppose that each feature in $X$ is associated with some scalar-valued meta-feature. We can represent these meta-feature values as a vector $m \in \mathbb{R}^p$, where the $i$-th entry in $m$ represents the value of our meta-feature corresponding to the $i$-th feature in the prediction problem. If, in some hypothetical scenario, we knew beforehand that the values in $m$ corresponded to each feature's global importance to the task at hand, a natural choice for $\Omega$ would be a penalty on the difference between our model's feature attributions and the values in $m$, giving us

$$\hat{f} = \underset{f \in \mathcal{F}}{\operatorname{argmin}} \frac{1}{n} \sum_{x,y} \mathcal{L}(f(x), y) + \lambda \sum_{x} \|\Phi(\theta_f, x) - m\|_1, \tag{1}$$

where $\|\cdot\|_1$ is the $L_1$ norm. We note that this function penalizes a difference between local attributions ($\Phi$), and global feature importance. While such a prior model may not make sense for some domains

(e.g. image classification tasks), previous work has empirically demonstrated benefits in prediction performance from regularizing attributions with a global prior in contexts where we might expect local attributions to be more correlated with a global prior (e.g. tabular biomedical datasets). Indeed, the results of Erion et al. [7] on gene expression data demonstrated improved generalization when regularizing deep networks such that their local attributions were close to pre-computed global importance values. Having such an $m$ that is known beforehand to be perfectly predictive of feature importance is unrealistic in most settings. However, suppose instead that we have some meta-feature matrix $M \in \mathbb{R}^{p \times k}$, corresponding to each of the $p$ features in the prediction problem having $k$ meta-features. If we were to assume that there is a linear relationship between a feature's meta-feature values and that feature's importance, we can modify Equation 1 to achieve

$$\hat{f} = \operatorname*{argmin}_{f \in \mathcal{F}} \frac{1}{n} \sum_{x,y} \mathcal{L}(f(x), y) + \lambda \sum_{x} \|\Phi(\theta_f, x) - \underbrace{\beta_1 M_{*,1} + \ldots + \beta_k M_{*,k}}_{\text{Linear attribution prior}}\|_1,$$

where $M_{*,i}$ refers to the $i$-th column of $M$, and $\boldsymbol{\beta} \in \mathbb{R}^k$ is a vector of hyperparameters that we can optimize over. Finding an optimal set of hyperparameters gives us a model that captures the meta-feature to feature importance relationship for a given problem. Assuming a linear relationship is restrictive, and likely fails to capture many of these relationships. Instead, we can take advantage of the universal function approximation capabilities of neural networks and replace our linear model with a deep one $\hat{g} \colon \mathbb{R}^k \to \mathbb{R}$ from some class of networks $\mathcal{G}$. We define such a model to be a deep attribution prior (DAPr). As either the number of meta-features $k$ increases or $\mathcal{G}$ becomes more complex, thereby leading to more model parameters, learning the parameters of a DAPr by treating them as hyperparameters becomes computationally prohibitive. Instead, we jointly learn a prediction model and DAPr pair $(\hat{f}, \hat{g})$ by optimizing the following objective:

$$(\hat{f}, \hat{g}) = \operatorname*{argmin}_{f \in \mathcal{F}, g \in \mathcal{G}} \frac{1}{n} \sum_{x,y} \mathcal{L}(f(x), y) + \lambda \sum_{x} \|\Phi(\theta_f, x) - G(M)\|_1,$$

where we define $G(M)$ to be the vector in $\mathbb{R}^p$ resulting from feeding each row of $M$ into $g$ (i.e., $G(M)$ is the vector of predicted importance values for all of our features based on each feature's meta-feature values). We are able to approximate a solution to this problem by alternating between one step of optimizing $\hat{f}$, and one step of optimizing $\hat{g}$ via a gradient descent-based optimization method. We find empirically that, so long as $\hat{f}$ and $\hat{g}$ change slowly enough, this procedure accomplishes the following two goals:

- Learn a relationship between provided meta-features and feature importance for a given prediction task.
- Bias prediction models to rely more heavily on the features deemed important by this learned meta-feature to feature importance relationship.

We show in Section 4 that introducing such bias in the training procedure improves the performance of deep neural network models in multiple settings where data is limited. Furthermore, we demonstrate in Section 5 that we can exploit these learned meta-feature to feature importance relationships to achieve new insights into our prediction models' behavior. For the experiments in the remainder of the text we use Expected Gradients (EG) [7] for our attribution method $\Phi$ because it has a low computational cost and resulted in superior performance as compared to other gradient-based attribution methods; we refer the reader to the Supplement for details on EG and additional experiments examining performance with other attribution methods.

## 4 Training with Deep Priors Improves Accuracy in Low-Data Settings

### 4.1 Two Moons Classification With Nuisance Features

We first evaluate the performance of multi-layer perceptrons (MLPs) with two hidden layers trained using the DAPr framework on the two moons with nuisance features task as introduced in Yamada et al. [41]. In particular we seek to understand whether our framework biases MLPs towards relying

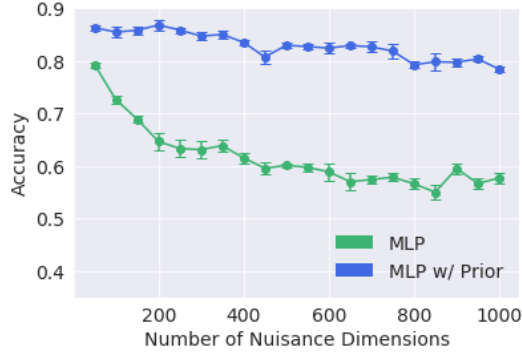

Figure 2: Classification accuracy (mean and standard error) for the two moons with nuisance features task of MLPs trained with and without the DAPr framework.

on informative features. In this experiment we construct a dataset based on two two-dimensional moon shape classes, concatenated with noisy features. For a given data point the first two coordinates $x_1, x_2$ are drawn by adding noise drawn from an $\mathcal{N}(0, 0.1)$ distribution to a point from one of two nested half circles. The remaining coordinates $x_3, \ldots, x_p$ are drawn from an $\mathcal{N}(0, 1)$ distribution. Our goal is to classify points as originating from one half-circle or the other.

For each of our experiments, we generate five datasets consisting of 1000 samples. For each dataset we use 20% of the dataset for model training, and divide the remaining points evenly into testing and validation sets, giving a final 20%-40%-40% train-test-validation split. We vary the number of nuisance features from 50 to 1000 in increments of 50. For a given number of features $p$ we construct a meta-feature matrix $M_{\mathrm{moons}} \in \mathbb{R}^{p \times 2}$, where the $i$-th row contains the $i$-th feature's mean and standard deviation. For our prior we train a linear model that attempts to predict feature importance based on that feature's mean and standard deviation. The number of units in the hidden layers of our MLPs depends on the number of features $p$; for a given $p$ the first hidden layer has $\lfloor p/2 \rfloor$ units, and the second has $\lfloor p/4 \rfloor$ units.

We optimize our models using Adam [14] with early stopping, and all hyperparameters are chosen based on performance on our validation sets (see Supplement for additional details). We report our results in Figure 2. We find that our MLPs trained with prior models using $M_{\mathrm{moons}}$ are far more robust to the addition of noisy features than MLPs that do not take advantage of meta-features. This result continues to hold even as the number of features in the dataset grows far beyond the number of training points. This phenomenon indicates that training an MLP with an appropriate prior model is able to bias the MLP towards learning meaningful relationships rather than overfitting to noise.

### 4.2 Alzheimer's Disease Biomarker Prediction

Alzheimer's disease (AD) is a slowly progressing neurodegenerative disorder affecting an estimated 5.7 million Americans [2]. Due to a rapidly aging population, this number is expected to increase to 13.8 million by 2050, and thus AD poses an increasing threat to healthcare systems. While AD has well-known biomarkers in the form of amyloid-$\beta$ protein deposits and misfolded tau proteins, the genetic drivers for these markers remain unclear [27].

In this experiment we explore how MLPs trained with the DAPr framework perform at using bulk RNA-seq data to predict amyloid-$\beta$ levels measured using immunohistochemistry. The data for this experiment comes from multiple datasets available on the Accelerating Medicines Partnership Alzheimer's Disease Project (AMP-AD) portal[1]; in particular we make use of the Adult Changes in Thought (ACT) [24], Mount Sinai Brain Bank (MSBB), and Religious Orders Study/Memory and Aging Project (ROSMAP) [1] datasets. All together these studies provide us with $n = 1742$ labelled gene expression samples. For prior information we gather a set of AD driver meta-features for each gene consisting of a binary variable indicating the presence of copy number variations, a second binary variable indicating whether the gene is a known regulator of other genes, methylation

Table 1: Performance (MSE $\pm$ SE) of MLPs trained with DAPr using disease-specific driver features vs. baseline models. * indicates the use of meta-features during training.

| Model | AD Biomarker Prediction | AML Drug Response |
|---|---|---|
| LASSO | $1.19 \pm 0.35$ | $1.031 \pm 0.051$ |
| MLP | $0.78 \pm 0.15$ | $0.938 \pm 0.106$ |
| MLP (L1 Reg.) | $0.74 \pm 0.12$ | $0.838 \pm 0.070$ |
| MLP (L2 Reg.) | $0.75 \pm 0.14$ | $0.871 \pm 0.075$ |
| MLP w/ Drivers* | $1.059 \pm 0.24$ | $1.158 \pm 0.117$ |
| MLP w/ DAPr (noise)* | $0.82 \pm 0.15$ | $0.915 \pm 0.079$ |
| MERGE [19]* | $1.07 \pm 0.21$ | $0.902 \pm 0.082$ |
| **MLP w/ DAPr (drivers)*** | **$0.67 \pm 0.13$** | **$0.786 \pm 0.065$** |

values, and node strength in a gene-gene interaction graph. We refer the reader to the Supplement for additional details on data collection and preprocessing. Using this prior information we construct a meta-feature matrix $M_{AD} \in \mathbb{R}^{p \times 4}$, where $p$ is the number of genes for which we have both expression data and driver feature values. In this experiment $p = 12,326$.

We compare our framework to multiple baselines and report our results in Table 1. To understand how incorporating meta-features into the training process improves model performance, we compare against LASSO [38], and MLPs trained without prior information. We also train MLPs with $L_1$ and $L_2$ regularization on their weights to illustrate how much of our performance gains comes from using prior knowledge specifically, as opposed to regularization in general. To demonstrate the benefits of using DAPr in particular to integrate meta-features into the training process, we compare against MLPs that naively make use of meta-features by simply treating them as additional input features. In addition, to confirm the informativeness of our specific meta-features, we compare training MLPs with DAPr models using AD driver features to MLPs also trained with DAPr models, but for which the prior information is composed solely of Gaussian noise. Finally, we compare our framework to the previous state of the art method for biasing model training using meta-features, MERGE [19], which biases a linear model's weights to match the output of a second linear model trained to predict weights from meta-feature values. All models are evaluated on five splits of the data into training, validation, and test sets. For each split we use 60% of the data for model training, 20% for validation, and 20% for testing. All prediction model MLPs have two hidden layers with 512 and 256 units respectively, and for our DAPr models we use MLPs with one hidden layer containing four units. For fairness we train all models using only the $p$ genes for which we have both expression data and meta-feature values. We refer the reader to the Supplement for further details on model training and evaluation.

We find that introducing meta-features into the training process can lead to better performance for both linear models and deep ones. However, care is required when using meta-features to train deep models; our naive method for training MLPs with meta-features leads to a degradation in performance. Our results also make clear the limitations of the linear models in MERGE, as even standard MLPs outperform it. Unsurprisingly, MLPs trained with DAPr models using noise meta-features perform slightly worse than standard MLPs. On the other hand, we find that MLPs trained with DAPr models using AD driver features see a 13.5% decrease in MSE as compared to standard MLPs. While we find minor boosts in performance by training MLPs with with $L_1$ and $L_2$ regularization, these improvements are far less than that of the MLPs with informative DAPr models. These results further indicate that DAPr models can bias MLPs towards learning meaningful nonlinear relationships not captured by simpler models even in high-dimensional, low sample size settings.

### 4.3 AML Drug Response Prediction

Acute myeloid leukemia (AML) is a form of blood cancer characterized by the rapid buildup of abnormal cells in the blood and bone marrow that interfere with the function of healthy blood cells. The disease has a very poor prognosis, leading to death in approximately 80% of patients, and it is the leading cause of leukemia-related deaths in the United States [40]. This phenomenon is likely due in large part to the heterogeneity of AML; different AML patients have varying responses to the same treatment regimens, and methods for better matching patients to drugs are in high demand.

In this experiment we explore how training an MLP with a DAPr model affects performance on an AML drug response prediction task. Our data comes from the Beat AML dataset, containing gene expression data and drug sensitivity measures for tumors from 572 patients [39]. In our analysis we focus on the drug Dasatinib, which had data from the most patients ($n = 217$). For meta-features we use the publicly available AML driver features from Lee et al. [19]: mutation frequencies, a binary variable indicating the presence of copy number variations, a second binary variable for whether the gene is a known regulator of other genes, gene expression hubness, and methylation values.

For this experiment we compare MLPs trained with DAPr models to the same baselines as in Section 4.2. We evaluate each model on five splits of the data. For each split, 80% of the data is used for training, while the remaining 20% is divided evenly into validation and test sets. We use the same prediction model MLP architecture as in Section 4.2. For our DAPr models we use MLPs with two hidden layers, of five and three units, to capture the relationship between meta-features and feature importance. We refer the reader to the Supplement for details on hyperparameter tuning.

Once again our results demonstrate that incorporating informative meta-features into the model training process can result in boosts in performance for both linear models and deep ones. While in this case MERGE performs on par with standard MLPs, our best performing models once again are MLPs trained with DAPr models.

## 5    Deep Attribution Priors Admit New Insights into Deep Models

While feature attribution methods provide users with a sense of the "relevant" features for a given prediction, they lack a way to contextualize the attribution values in a human-interpretable way. This shortcoming can lead to a false sense of security when employing such methods, even though previous work has demonstrated their potential to produce explanations known to be nonsensical based on prior human knowledge [42, 13, 10]. However, we can take a step towards alleviating this problem by probing a DAPr model $\hat{g}$ to understand how meta-features drive changes in its predicted global feature importance values. In the case where $\hat{g}$ is a linear model, we can simply use the model's weight coefficients to get an explanation for a given predicted importance value. When $\hat{g}$ is a deep model, uncovering such explanations is not as straightforward. However, in this section, using the AML drug response prediction problem from Section 4.3 as a case study, we demonstrate multiple methods for probing DAPr models to obtain insights into their predictions. Using such methods we find that our DAPr model from Section 4.3 independently learns meta-feature to gene importance relationships that agree with prior biological knowledge.

### 5.1    Attribution Explanations For Understanding Meta-Feature to Gene Relationships

Given a potentially complex DAPr model $\hat{g}$ and vector of meta-feature values $m_i$ for the $i$-th feature in a prediction problem, we would like to understand how the values of $m_i$ relate to that feature's predicted global importance value $\hat{g}(m_i)$. To do so we can apply an attribution method $\Phi$ to $\hat{g}$, thereby generating a second order explanation $\Phi(\theta_{\hat{g}}, m_i)$, explaining the $i$-th feature's predicted global importance in terms of human-interpretable meta-features. We apply EG as our $\Phi$ to the DAPr model from Section 4.3, and visualize our results in Figure 3.

Remarkably, we find that our DAPr model rediscovers relationships in line with prior biological knowledge. We can see in Figure 3 that high predicted gene importance values are explained mostly by expression hubness. This trend is consistent with prior knowledge, as expression hubness has been suspected to drive events in cancer [21]. Furthermore, we observe that, for a small number of genes, mutation appears as a noticeable factor in their importance explanations. This phenomenon also agrees with prior knowledge, as mutations in the genes CEPBA, FLT3, and ELF4 are suspected to play a role in the variability of drug response between AML patients [20, 8, 34, 6, 36]. We also note that our framework, on its own, selected for relevant meta-features among those provided. This behavior validates that DAPr can reduce the domain knowledge requirement of the original attribution prior framework; rather than needing a specific meta-feature to feature important relationship up front, DAPr can select for relevant meta-features out of those provided.

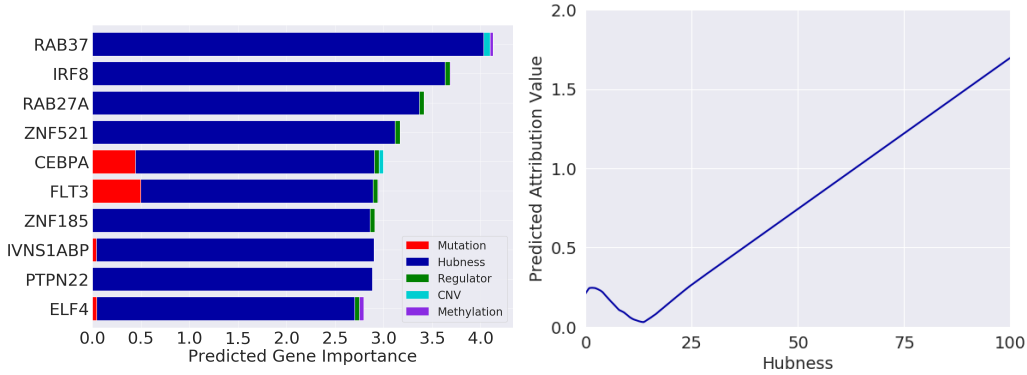

Figure 3: (Left) Explanations of feature importance for most important genes as ranked by absolute predicted importance. Bar color proportional to absolute value of that meta-feature's contribution to the predicted importance value as determined by EG. (Right) Deep attribution prior PDP for expression hubness.

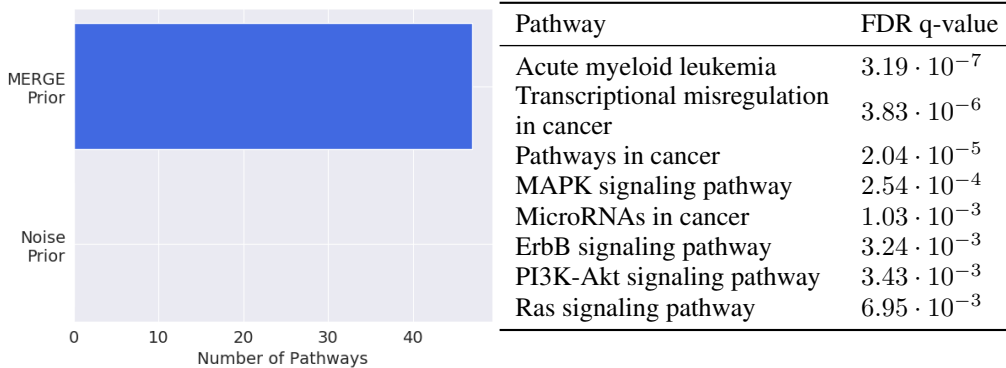

| Pathway | FDR q-value |
|---|---|
| Acute myeloid leukemia | $3.19 \cdot 10^{-7}$ |
| Transcriptional misregulation in cancer | $3.83 \cdot 10^{-6}$ |
| Pathways in cancer | $2.04 \cdot 10^{-5}$ |
| MAPK signaling pathway | $2.54 \cdot 10^{-4}$ |
| MicroRNAs in cancer | $1.03 \cdot 10^{-3}$ |
| ErbB signaling pathway | $3.24 \cdot 10^{-3}$ |
| PI3K-Akt signaling pathway | $3.43 \cdot 10^{-3}$ |
| Ras signaling pathway | $6.95 \cdot 10^{-3}$ |

Figure 4: (Left) Number of pathways captured by DAPr trained using AML driver features vs. DAPr trained with noise. (Right) Sample of captured pathways believed to be relevant to AML.

## 5.2 Partial Dependence Plots Capture Nonlinear Meta-Feature to Feature Importance Relationships

We can further explore the nature of these learned meta-feature to feature importance relationships by constructing partial dependence plots [9] to visualize the marginal effect of particular meta-features on predicted global importance values. We display the results of doing so for expression hubness in Figure 3. We find that our DAPr model captures a non-linear relationship between hubness and predicted global importance. This relationship broadly agrees with prior knowledge; after an initial buildup increases in hubness are strongly associated with a gene's relevance to drug response. Our other meta-features also exhibited nonlinear relationships with predicted feature importance, and we refer the reader to the Supplement for additional figures. The nonlinearities captured by our DAPr model indicate that more complex models may better capture the relationship between a gene's meta-features and its potential to drive events in AML than e.g. the linear models used by MERGE.

## 5.3 Deep Attribution Priors Capture Relevant Gene Pathways

To further confirm that the relationships captured by our DAPr model match biological intuition, we perform Gene Set Enrichment Analysis [35] to see if the top genes as ranked by our prior were enriched for membership in any biological pathways. For comparison we use the number of pathways captured by a DAPr model trained on noise as a baseline. In our analysis we use the top 200 genes as ranked by both prior models, and we use the Enrichr [16] library to check for membership in the Kyoto Encyclopedia of Genes and Genomes [12] 2019 pathways list. While the top genes as ranked by our noise DAPr model are not significantly enriched for membership in *any* pathways

after FDR correction, our AML driver DAPr model captures many pathways, of which multiple are already believed to be associated with AML [23, 32, 25, 26]. This result further indicates that our deep attribution prior is capturing meaningful meta-feature to gene importance relationships.

# 6  Discussion

In this work we introduce the DAPr framework for biasing the training of neural networks by incorporating prior information about the features used for a prediction task. Unlike other feature attribution based penalty methods, our framework merely requires the presence of prior information in the form of meta-features, rather than specific rules hand-crafted by a domain expert. In our experiments we find that jointly learning prediction models and DAPr models leads to increased performance on prediction tasks in settings with limited data. Furthermore, we demonstrate that the DAPr framework admits new methods for understanding deep models. Using such methods we demonstrate that our prior models independently learn meta-feature to feature importance relationships that agree with prior knowledge. The DAPr framework provides a broadly applicable method for incorporating prior knowledge in the form of meta-features into the training of deep neural networks, and we believe that it is a valuable tool for learning in low sample size, trust-critical domains.

## Broader Impact

DAPr can be applied to a wide variety of problems for which prior knowledge is available about a dataset's individual features. In our work we focus on applying our method to a synthetic dataset and two real-world medical datasets, though it should be easily extendable to other problem domains.

As discussed in the introduction, a major barrier to the adoption of modern machine learning techniques in real-world settings is that of *trust*. In high-stakes domains, such as medicine, practitioners are wary of replacing human judgement with that of black box algorithms, even if the black box consistently outperforms the human in controlled experiments. This concern is well-founded, as many high-performing systems developed in research environments have been found to overfit to quirks in a particular dataset, rather than learn more generalizable patterns. In our work we demonstrate that the DAPr framework does help deep networks generalize to our test sets when sample sizes are limited. While these results are encouraging, debugging model behavior in the real world, where data cannot simply be divided into training and test sets, is a more challenging problem. Feature attribution methods are one potential avenue for debugging models; however, while it may be easy to tell from a set of attributions if e.g. an image model is overfitting to noise, it would be much more difficult for a human to determine that a model trained on gene expression data was learning erroneous patterns simply by looking at attributions for individual genes. By learning to explain a given feature's global importance using meta-features, we believe that DAPr can provide meaningful insights into model behavior that can help practitioners debug their models and potentially deploy them in real-world settings.

Nevertheless, we recognize the potential downsides with the adoption of complex machine learning interpretability tools. Previous results have demonstrated that interpretability systems can in fact lead users to have *too much* trust in models when a healthy dose of skepticism would be more appropriate. More research is needed to understand how higher-order explanation tools like DAPr influence user behavior to determine directions for future work.

## Acknowledgments and Disclosure of Funding

This work was supported by funding from the National Science Foundation [CAREER DBI-1552309 and DBI-1759487], American Cancer Society [127332-RSG-15-097-01-TBG], and National Institutes of Health [R35 GM 128638 and R01 NIA AG 061132]. The results published in this work are in part based on data obtained from the Alzheimer's Disease Knowledge Portal (`https://adknowledgeportal.synapse.org/`) operated by Sage Bionetworks. The results published here are partially based upon data generated by the Cancer Target Discovery and Development (CTD2) Network (`https://ocg.cancer.gov/programs/ctd2/data-portal`) established by the National Cancer Institute's Office of Cancer Genomics. We thank the members of the Lee Lab and the anonymous NeurIPS reviewers for their feedback that significantly improved this work.

## Footnotes

[1]`https://adknowledgeportal.synapse.org`

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
