[Supplementary Material]

# Supplement to Learning Deep Attribution Priors Based On Prior Knowledge

## 1   Model Implementations and Hyperparameter Tuning

**LASSO:** In our experiments we used the scikit-learn [10] implementation of the LASSO algorithm [12]. For each experiment we sampled $\alpha$ values from the range $(0, 1]$, and we chose the value of $\alpha$ with the smallest MSE on our validation set for evaluation on the test set.

**MERGE:** All linear models were implemented using PyTorch. We optimized these models using ADAM. To choose learning rates for our prediction and prior models, we performed grid searches over the range $10^{-6}$ to $10^{-3}$ for both parameters.

**Neural Networks:** All neural network models were implemented using the PyTorch framework [9]. We used an Nvidia GTX 1080 Ti GPU for training.

For the noisy two moons task, all neural networks consisted of two hidden layers, with the number of hidden units varying depending on the dimensionality of the experiment. For a given number of hidden layers $p$, our first hidden layer had $\lfloor p/2 \rfloor$ units, and the second had $\lfloor p/4 \rfloor$ units. All prediction networks were trained using Adam optimizers [5] to minimize the cross entropy loss plus an $L_1$ penalty on feature attribution differences as described in the main text.

For our experiments on biological datasets, all neural network models used for prediction consisted of two hidden layers with 512 and 256 hidden units respectively. When constructing our learned attribution prior networks, we experimented with multiple architectures. In our experiments we found that a network with a single hidden layer of 4 units worked best for the AD prediction task, while a network with two hidden layers of 5 and 3 units respectively performed best for the AML prediction task. All prediction models were trained using Adam optimizers to minimize MSE loss plus the $L_1$ penalty on feature attribution differences as described in the main text. The number of training epochs for each network was determined by early stopping. In our experiments we used a patience parameter of 20. Our learning rates were selected by tuning from the range $10^{-6}$ to $10^{-3}$ based on performance on our validation set. For our baseline MLPs trained with L1/L2 regularization on their weights, we considered values of $\lambda$ (i.e., regularization strength) between $5 \cdot 10^{-3}$ and $5 \cdot 10^{-5}$.

We found that setting the DAPr regularization strength $\lambda$ to $1/m$, where $m$ is the size of a minibatch, resulted in a good balance between the terms in our loss function across our datasets. As such, we used this value for all experiments.

## 2   Choice of Attribution Method

As discussed in the main text, one of the hyperparameter choices required when using the DAPr framework is the specific attribution method $\Phi$. For our experiments we ended up using Expected Gradients (EG) [2], due to both its computational efficiency as well as its performance.

EG is an extension of Integrated Gradients (IG) [11] designed to avoid hyperparameter choices required by IG. IG computes feature attributions by comparing a model's prediction with the prediction that would be made given some baseline input that represents a lack of information. For example, in

image classification tasks an image of all black pixels is often taken as a baseline. However, in many other domains it is unclear what a zero-information baseline would be.

EG avoids this issue by integrating over the dataset to find a baseline value for a feature, rather than specifying the baseline by hand. For a given model $f$, the IG value for the $i$-th input feature is defined as

$$\text{IG}_i(x) := (x_i - x'_i) \times \int_{\alpha=0}^{1} \frac{\partial f(x'_i + \alpha \times (x - x'))}{\partial x_i} d\alpha,$$

where $x$ is a target input and $x'$ is a baseline input. EG avoids specifying $x'$ by taking a second integral over the dataset, giving the equation

$$\text{EG}_i(x) := \int_{x'} \left( (x_i - x'_i) \times \int_{\alpha=0}^{1} \frac{\partial f(x'_i + \alpha \times (x - x'))}{\partial x_i} d\alpha \right) p_D(x') dx',$$

where $D$ is the distribution of our dataset. Computing such an integral directly is computationally intractable. However, this equation can be rewritten in terms of expectations giving

$$\text{EG}(x_i) = \mathop{\mathbb{E}}_{\substack{x' \sim D, \\ \alpha \sim U(0,1)}} \left[ (x_i - x'_i) \times \frac{\partial f(x'_i + \alpha \times (x - x'))}{\partial x_i} \right].$$

This expectation can be approximated by sampling $(x', \alpha)$ pairs from $D$ and $U(0, 1)$, computing the value inside the expectation for each pair, and then averaging over samples. In Erion et al. [2] the authors found that, for neural network models trained using some form of batch gradient descent, sampling only one $(x', \alpha)$ pair per mini-batch during model training was sufficient to regularize a model's attributions using EG. As such, using EG we can compute $\Phi$ using only one additional gradient call per batch.

We also found that EG led to the best performance for models trained using the DAPr framework. We report results from earlier exploratory experiments on both the Alzheimer's and AML datasets below.

Table 1: Performance (MSE $\pm$ SE) of MLPs trained with DAPr using different attribution methods $\Phi$.

| Model | AD Biomarker Prediction | AML Drug Response |
|---|---|---|
| Input Gradients | $0.779 \pm 0.134$ | $0.946 \pm 0.090$ |
| Input * Gradients | $0.772 \pm 0.142$ | $0.898 \pm 0.103$ |
| Integrated Gradients | $0.786 \pm 0.159$ | $0.844 \pm 0.080$ |
| **Expected Gradients** | **$0.67 \pm 0.13$** | **$0.787 \pm 0.0655$** |

On both our biological datasets, EG led to the best performance using the DAPr framework.

## 3 Two Moons with Nuisance Features

### 3.1 Final Hyperparameter Values

**MLP (no prior):**

- Learning rate: $5 \cdot 10^{-4}$

**MLP (with prior):**

- Learning rate: $5 \cdot 10^{-4}$
- Prior learning rate: $1 \cdot 10^{-3}$

# 4 AML Drug Response Experiments

## 4.1 Final Hyperparameter Values

**LASSO:**

- $\alpha$: 0.54

**MERGE:**

- Learning rate: $5 \cdot 10^{-5}$
- Prior learning rate: $5 \cdot 10^{-5}$

**MLP (no prior):**

- Learning rate: $1 \cdot 10^{-3}$

**MLP (noise prior):**

- Learning rate: $1 \cdot 10^{-3}$
- Prior learning rate: $5 \cdot 10^{-5}$

**MLP (AML driver prior):**

- Learning rate: $1 \cdot 10^{-3}$
- Prior learning rate: $5 \cdot 10^{-5}$

**MLP with meta-features (naive)**

- Learning rate: $1 \cdot 10^{-3}$

**MLP (L1 regularization)**

- Learning rate: $1 \cdot 10^{-3}$
- $\lambda$ (L1 regularization strength): $1 \cdot 10^{-4}$

**MLP (L2 regularization)**

- Learning rate: $1 \cdot 10^{-3}$
- $\lambda$ (L2 regularization strength): $5 \cdot 10^{-4}$

## 4.2 Data Preprocessing

Due to the potential for noise and batch effects, it is necessary to preprocess RNA-seq gene expression data in order to ensure a quality signal. For our AML drug response dataset, we preprocessed our RNA-seq data as follows

1. Raw transcript counts were converted to fragments per kilobase of exon model per million mapped reads (FPKM). FPKM normalizes the counts for different RNA lengths and for the total number of reads, and as such is more reflective of the molar amount of a transcript in the original sample than raw counts [8]. FPKM is calculated using the following formula

$$FPKM = \frac{X_i \cdot 10^9}{N l_i}$$

   Where $X_i$ is the raw count for a transcript, $l_i$ is the effective length of the transcript, and $N$ is the total number of counts.

2. Non-protein coding transcripts were removed from the dataset.

3. Transcripts not meaningfully observed in the dataset ($> 70\%$ measurements equal to 0) were removed.

4. The data was $\log_2$ transformed.

5. Each transcript was standardized across all samples, so that the mean for the transcript was equal to 0 and the variance was equal to one. I.e.,

$$X_i' = \frac{X_i - \mu_i}{\sigma_i}$$

We also scaled Dasatinib IC50 values to have zero mean and unit variance.

## 4.3 Gene Meta-Features

We downloaded the MERGE (Mutation, Expression hubness, Regulator, Genomic copy number variation, and mEthylation) prior feature data from the Supplement of [6]. Here we provide a brief description of how each of the prior features were originally calculated.

- *Expression hubness*: SPARROW (SPARse selected expRessiOn regulators identified With penalized regression), is a computational method for estimating a gene's hubness purely based on expression data from cancer patients. SPARROW employs a sparse statistical model in which each gene's expression level is modeled as a linear combination of a small set of other genes (i.e., a sparse basis). A gene's hubness is determined based on how often the gene is chosen in the sparse basis for any other gene. To use gene hubness as a MERGE feature, the the R data object (.rda) containing the SPARROW results for AML from `http://sparrow-35leelab.cs.washington.edu/data` was downloaded. The `sparrow1` scores (the number of downstream genes) from the `basesFreq` object were then used as the expression hubness feature.

- *Mutation*: Significance measures of mutation frequencies for each gene measured in the MutSig2CV Analysis of the AML study from TCGA (`http://firebrowse.org/?cohort=LAML`) were obtained. A p-value was assigned to each gene measuring the statistical significance that the gene had mutated more often than expected by chance. $-\log_{10}$ (p-value) was used as the feature value.

- *Genomic copy number variation*: CNV measures (`gdac.broadinstitute.org_LAML-10TB.CopyNumber_Gistic2.Level_4.2015082100.0.0.tar.gz`) were downloaded from `http://gdac.broadinstitute.org/runs/analyses__2015_08_21/data/LAML/20150821/`. The file `all_data_by_genes.txt` found in the tar.gz file was then used to assign 1 (CNV) or 0 (no CNV) to each gene. The CNV feature of a gene was set to 1 if the gene was amplified or deleted by at least .05 in at least 20 of 191 patients ($\sim 10\%$), and to 0 otherwise.

- *Regulator*: It is likely that genes known to regulate other genes are more reliable molecular markers for therapeutic response than those that are not. As such, list of genes known to have regulatory roles, including transcription factors, chromatin remodelers and signal transduction genes, was constructed based on gene annotation databases [3]. Based on this list, a binary feature for each gene was generated by assigning 1 if the gene was on the list and 0 otherwise.

- *Methylation*: DNA methylation measures (`gdac.broadinstitute.org_LAML.Methylation_Preprocess.Level_3.2015110100.0.0.tar.gz`) were obtained from `http://gdac.broadinstitute.org/runs/stddata__2015_11_01/data/LAML/20151101/`. The file `LAML.meth.by_mean.data.txt` in this tar.gz was then used to obtain the average methylation levels for each gene across all patients.

# 5 Alzheimer's Disease Experiments

## 5.1 Data Access

MSBB neuropathology data was obtained from the AMP-AD Knowledge Portal of Sage Bionetworks through https://www.synapse.org/ with Synapse ID syn6101474. ROSMAP RNA-Seq data and MSBB RNA-Seq were similarly obtained via Synapse IDs syn3505732 and syn7391833, respectively.

## 5.2 Final Hyperparamter Values

**LASSO:**

- $\alpha$: 0.36

**MERGE:**

- Learning rate: $5 \cdot 10^{-6}$
- Prior learning rate: $5 \cdot 10^{-6}$

**MLP (no prior):**

- Learning rate: $1 \cdot 10^{-5}$

**MLP (noise prior):**

- Learning rate: $1 \cdot 10^{-5}$
- Prior learning rate: $1 \cdot 10^{-4}$

**MLP (AD driver prior):**

- Learning rate: $1 \cdot 10^{-5}$
- Prior learning rate: $1 \cdot 10^{-4}$

**MLP with meta-features (naive)**

- Learning rate: $1 \cdot 10^{-5}$

**MLP (L1 regularization)**

- Learning rate: $1 \cdot 10^{-5}$
- $\lambda$ (L1 regularization strength): $1 \cdot 10^{-3}$

**MLP (L2 regularization)**

- Learning rate: $1 \cdot 10^{-5}$
- $\lambda$ (L2 regularization strength): $1 \cdot 10^{-3}$

## 5.3 Data Preprocessing

RNA-seq values were preprocessed using the same procedure as in our AML experiments as described in Section 4.2. We also scaled our amyloid-$\beta$ values to have zero mean and unit variance.

## 5.4 Gene Meta-Features

- *Genomic copy number variation*: Values for CNV were reused from the AML driver feature set as described in Section 4.3.
- *Regulator*: Known regulator values were reused from the AML driver feature set as described in Section 4.3.
- *Methylation*: DNA methylation measures (`GSE80970_family.soft.gz`) were obtained from `ftp://ftp.ncbi.nlm.nih.gov/geo/series/GSE80nnn/GSE80970/soft/`. The file `HumanMethylation450_15017482_v1-2.csv` was then used to obtain average methylation levels for each gene across all patients.
- *Strength*: For a weighted graph, Barrat et. al define in Barrat et al. [1] the strength of the $i$-th node as the sum of the weights of all edges attached to the node. For the genes in our AD RNA-seq data, we compute their strength using the edges in the HumanBase [4] brain tissue graph available at `https://hb.flatironinstitute.org/download`. For our experiment we chose the "top edges" version of the graph.

# 6 Partial Dependence Plots

As discussed in the main text, we produced partial dependence plots to understand our DAPr model's behavior for the AML prediction task. We produced plots for three out of the five features (hubness, mutation, and methylation); the other two (copy number variation, known regulator) were binary variables and so we did not feel it appropriate to apply PDPs to analyze them.

Figure 1: PDP for expression hubness

The first plot we produce (also seen in the main text) is for expression hubness (Figure 1). The nonlinear trend seen here matches with prior biological knowledge [6]; after an initial buildup, increases in expression hubness are consistently associated with higher feature importance. Our PDP for mutation (Figure 2, left) reveals a similar trend that again agrees with prior knowledge. For methylation (Figure 2, right) we see a different nonlinear trend. Rather than seeing an increase in predicted importance after an intial buildup, we instead see a decrease. This trend also agrees with prior knowledge. It is well-known that methylation inhibits gene expression [7], so we would expect that expression values for genes with higher methylation values are less informative for our prediction task.

Figure 2: PDP for mutation (left) and methylation (right)