[Reviews · NeurIPS 2020]

Review 1

Summary and Contributions: This paper proposes a novel feature attribution method called deep attribution prior (DAPr) which used prior knowledge on what features could be more useful in predicting tasks. Interpretations of deep learning models are essential in understanding the importance of features, especially in critical tasks such as medicine. This paper introduces a deep attribution method that can use any known relationships between features and meta-features and can force the model to use more relevant features for the prediction.

Strengths: The main strength of this paper is that it has the capability to take into account any prior knowledge about the associations of features and and their meta-features by adding a guiding loss to the model prediction loss as opposed to any typical L1 or L2 regularization loss. This leads to better performance than other feature attribution methods when the number of data is small and there is some prior knowledge to be used for model interpretations. Using deep attribution priors also makes the predictions more robust against noise in the datasets.

Weaknesses: One weakness of the paper is that it assumes the prior knowledge in terms of an association between the features and meta-features. This could be a restrictive assumption as we could have other form of prior knowledge which could not be represented by a feature-meta-feature association matrix. Another weakness is that there is not enough technical novelty in the work.

Correctness: Yes

Clarity: Yes

Relation to Prior Work: Yes

Reproducibility: Yes

Additional Feedback: The idea of using prior knowledge in feature attribution methods is promising. However, the form of prior knowledge could be different from the meta-feature matrix assumed in this paper. For example, we could have prior knowledge in terms of some gene pathways or networks or graphs. How can this method deal with the prior knowledge when it is in different format than meta-feature matrices? Also, it is not clear how to design the meta-feature matrices M. The selection of meta-features and their numbers should be determined by some experts which is not an easy job. Furthermore, the technical novelty in the paper is not that much and it seems similar to Expected Gradients (EG)'s paper. The response has been read.


Review 2

Summary and Contributions: The paper proposes DAPr, a framework to regularize attributions made by a machine learning model to align with prior information. Given some kind of prior information, a deep model learns to approximate the importance of each meta-feature. ** I have read the rebuttal. My score remains unchanged

Strengths: The problem that this paper is working on is highly relevant to the community. Learning how to incorporate prior information into a model is a topic of active research. The idea is novel and well-motivated. The paper is overall well-written.

Weaknesses: The evaluation of the framework is relatively weak. For 4.1, the model is not compared to any prior work, despite f.e. Ross 2017 (find-another-explanation) providing a suitable baseline. In 4.2, the authors mention excluding features for which no attribution prior was available. How does a model with all genes compare?

Correctness: To the best of my knowledge, the claims and methods are correct. In table 1, there is significant overlap between the mean \pm SE between methods. I would like to see a statistical test to find out if the DAPr model in fact performs better than an ML with L1 reg instead of just bolding the method with the lowest mean error.

Clarity: In general the paper is well written and is easy to follow through. Section 4.2 was confusing to read, though this may be in part due to unfamiliarity with the domain.

Relation to Prior Work: The related works section is relatively short and does not sufficiently set this work into context with prior work. As an example, Ross 2017 which also provides an approach to learn attribution penalization without prior information is only mentioned in passing.

Reproducibility: Yes

Additional Feedback: In general I like this paper and believe it would be suitable for NeuRIPS, particularly if the issues mentioned above are adressed. In the equation in l.109 beta Is not included in the argmin The meta-features are vital for understanding the paper.They are currently not explained well in a way that facilitates understanding of the later text. For me personally the introduction of m first as a vector with a 1:1 relation to the features and then as the matrix M did not help with this.


Review 3

Summary and Contributions: This paper proposes the deep attribution prior framework (DAPr) for regularizing attributions to align with "meta-feature" information about each of the observed features. The proposed method jointly trains a prediction model and a prior model that predicts the attribution of the prediction model given meta-features. ********** I thank the authors for including an additional baseline. Looking at the author's response, I think they did a reasonable job of addressing my comments. However, I do agree with the concerns brought up by R4 that enforcing the same attribution prior across all samples is undesirable (especially if the "true" attribution for a subset of the points does not align with the majority class) and that the careful tuning of lambda (and the difficulty to converge) seems to still be an issue. As such, I will bump my score from a 4 to a 5.

Strengths: - The paper provides a nice way to incorporate meta-features (additional information about features into the learning procedure). Instead of regularizing for the "ideal" attribution for a particular data point, the framework finds the predicted attribution for a particular point given the meta-features. - The use of meta-features is quite clever, and the joint training setup (of the prior and prediction models) seems sensible. - The fact that DAPr learns "meta-features to gene importance relationships that agree with prior biological knowledge" is quite amazing! - The code and experimentation seem sound. - The idea of meta-features is interesting given recent work on incorportating human priors into models [29,30,7].

Weaknesses: - Giving a concrete example of the meta-features (in more detail than the movie example on line 42) would help ground the paper's contribution from the start. It would be nice to have this as a running example throughout the paper; perhaps pull the AD or AML example to the introduction. - The need to craft meaningful meta-features (drivers) seems burdensome, though it's definitely more realistic than providing exact attributions or binary relevance (as in RRR). The fact that noise priors don't help is not promising when you don't have hand-crafted meta-features. Also, what happens when you only have meta-features for one feature? Can your method handle partial meta-features (<k meta-features for some of the p features) or handle class-specific meta-features? Some Other Questions - It looks like DAPr performs slightly better than regularized MLPs. Can you please report what happens when DAPr is run on an MLP with L1 or L2 Reg? Do you achieve even lower MSE? - Also, can you discuss how much the parameters vary between DAPr trained prediction models and regularized models? It would be nice to see an analysis on what type of regularization the meta-features are inducing on the regularization procedure. Have the prediction models' parameters learned anything meaningful from the meta-features? - Also, how senstive is your prediction model to the capacity you give the prior model? Is there an optimal capacity for how much to parametrize the prior model? I see you picked an MLP with one hidden layer with 4 units for AD. Was this an arbitrary choice?

Correctness: The DAPr method seems correct and the empirical experiments are well-documented.

Clarity: The paper reads clearly; there were no obvious errors that detracted from the reading experience.

Relation to Prior Work: The relation to prior work is thorough; however, it would have been nice to constrast the approaches of [29, 30] with DAPr and EG. Most of the machinery for DAPr seems to build off of EG [7].

Reproducibility: Yes

Additional Feedback:


Review 4

Summary and Contributions: This work presents a method, which trains a neural network to rely on only the most appropriate features (instead of noise) when making its prediction. To accomplish this, at training, the feature attributions/importances are computed for each data point. Along with the predictive loss, the model is penalized if the attributions differ from an expected global feature importance vector (i.e. the expected importance of each feature in general, shared across all input data points). The global feature importance vector is learned by a separate neural network (termed a “DAPr” model); specifically, each feature is associated with a small set of “meta-features”, which describe that feature in a human interpretable way, and this neural network learns the relationship between the meta-features of a feature, and how important that feature should be for prediction in general. The authors apply this method to the synthetic “two-moons” classification task, an Alzheimer’s predictor based on gene expression, and an AML drug response predictor based on gene expression. The authors subsequently show that using their method, models are able to achieve higher predictive performance. Furthermore, the trained DAPr model can reveal the driving meta-features behind why a feature is important, a relationship which can be complex and nonlinear. ************************* UPDATE AFTER AUTHOR RESPONSE My biggest concern remains that the learned attribution prior rewards/penalizes importance of the same exact features across all training examples (both positives and negatives). The assumption that all inputs should be encouraged to have the same attributions isn't correct in general, and this almost certainly includes the real-world datasets used in this paper. Different training examples have different features that are driving it to be positive or negative. For example, diseases like Alzheimer's or phenotypes like drug response are often driven by different genes in different people. Of course, the attribution prior isn't _forcing_ every training example to have the same attributions, but it is directing the model to do so as much as possible (tuned by lambda). I think this faulty assumption can lead to several downstream issues that are sometimes evident in the paper: 1. The models might be very difficult to train/converge, perhaps evident by the abnormally large patience used for early stopping. It's likely that the model is trying to learn to predict Alzheimer's (to use the paper's example here), and to do so, it needs to rely on different genes for different people; but every so often, the attribution prior is pulling the model back to always rely on the same exact features for each example (which the model does not want to do, because then it would incorrectly classify some examples). 2. The "important features" that the DAPr framework identifies will very likely have poor recall. Because all input examples are encouraged to have the same attributions, the network will probably only pick up on the features that are most _commonly_ informative. Input examples that have a different set of informative features (compared to the majority) might be disproportionately misclassified, and these less-common informative features will not be picked up by the global feature importance vector. I'm a bit worried that (without adequate explanation), this method might be used in the field to identify important features in a dataset, while quietly failing to pick up on all the features that are important for many of the training examples (which are still in the minority of the training set). I would be much more comfortable if the story were repackaged as the following: - Novel contributions: 1) learning expected global attributions from metafeatures using a neural network instead of linear regression; 2) applying the global attributions as an attribution prior - Main result: improved performance on predictions when training with the global attribution prior, possibly because the model is directed to ignore features that are universally non-informative (e.g. Alzheimer's is probably never related to the SRY gene) - Caveats: 1) important features might be missed, especially if they are in the minority of training examples; 2) the attribution prior competes with the main loss rather than working cooperatively, so careful tuning of lambda is required ********************************************

Strengths: Attribution priors are a burgeoning field within interpretable deep learning, and this work attempts to alleviate some of the limitations of traditional feature attribution priors—specifically, it is sometimes difficult to know ahead of time what features are important for a prediction, which would prevent us from being able to penalize the model for putting attribution on the wrong features. This work asks whether we can learn which features are important for a prediction using a separate neural network trained on cleverly chosen meta-features. This is a thought-provoking idea that is explored through empirical experiments. While the idea of learning feature importance from metafeatures is not entirely new (and the authors briefly mention this), learning this relationship using a neural network, and then subsequently introducing the result as an attribution prior, is—to the best of my knowledge—novel. The authors also demonstrate the advantage of being able to learn non-linear relationships between meta-features and feature importance, shown by an increase in performance, and a suggested improvement in interpretability. To demonstrate this claim, the authors show three experiments on both synthetic and real data, and the improvements in performance hold across all three. Empirically, this is strong evidence of a good contribution. Finally, with proper selection of human-interpretable meta-features, this method can help explain why a feature is important for the prediction, in an interpretable way. As a result, this can help increase the trust that humans have for certain models, since they can potentially confirm that the model is relying on the right features for the right reasons.

Weaknesses: A huge issue with this work is that this method assumes every input data point should have the same attributions. The attribution prior penalizes attributions against a (learned) global set of feature importances that are shared across all training examples. This encourages the prediction model to rely on the same features in the same proportions, regardless of what the input is. In general, for most prediction tasks, this is not true! Which features are important often depends very heavily on the values of the features. That is, different inputs can (and should) have different attributions. This could seriously impair the “recall” of important features. For example, in the Alzheimer’s dataset, imagine that most positive cases had abnormalities in a few genes (geneA, geneB, or geneC), but there is a very small number of people who had abnormalities in a completely different gene (geneD). By forcing every patient to have the same set of attributions, it is very possible (if not likely) that the model learns to ignore geneD completely, even though it is highly informative for a few patients. In a similar vein, it seems inappropriate to apply the same attribution requirement to both positives and negatives for classification tasks like the Alzheimer’s dataset. In general, negatives do not have the same attributions as positives, even after taking the absolute value of the attributions. In general, negatives are characterized by the absence of positive-driving features, not the presence of negativedriving features. Even if the negatives were driven by a few select features, it is unlikely that they are the same as the positive-driving features. Again, using a global feature importance for all training examples, positives and negatives, is inappropriate. It would be informative to look at the attributions of negative examples, and see if all the attributions are allocated to the same features. Additionally, trying to force every input to have the same feature attributions severely limits the ability to identify meaningful feature attributions from the predictive network. In fact, if the DAPr method is working “correctly”, it would ablate meaningful feature attributions in the different training examples. Furthermore, this method and its advantages are limited to instances where we can identify a set of informative meta-features that are interpretable to humans. As shown, the selection of meta-features is very important for performance and interpretability. Furthermore, analyses like the one in section 5.1 are obviously limited to the meta-features that were selected at the beginning (i.e. you can only interpret what you include). Coupled with the requirement that all training examples have the similar feature attributions, this severely limits the applicability of this method. In addition to the requirement for domain-specific knowledge to identify a good set of meta-features, some domains are just not conducive to these kinds of meta-features. For example, this method could not be reasonably applied to something like image classification, as the only meta-features that can describe pixels in a meaningful way would be relating to the location of the pixels. On the subject of training stability, I suspect that training these models is challenging, and convergence is difficult to attain. In particular, the authors note they trained the models with an early-stopping patience of 20, which is an abnormally large number of epochs to wait. This suggests that the training might be very finicky, and might even be oscillating. Given the other limitations listed above, this is concerning, because the model might be trying to put importance on certain features for some training examples, and shifting importance to other features for other training examples. To examine this concern, the authors should include training/validation loss curves, and a visualization of how the global feature importances changes over the training run. On the Alzheimer’s dataset, it’s rather noticeable that no analyses on interpretability were offered. Even though the Alzheimer’s task and AML drug response task had the same features (i.e. gene expression) and nearly identical meta-features, interpretation of the driving meta-features and global feature importances were only examined for the AML task. This is concerning, because it casts doubt on the one of the most important claims in the paper: that training with a DAPr model improves the interpretability of the features. Of the three experiments shown, the synthetic two-moons experiment is designed to be trivial for DAPr (although the results of interpretability on this task should be shown, as well), and only one of the real-data experiments has interpretability results. To sufficiently demonstrate that this claim holds, the authors will need to show that the most important genes and their driving meta-features for the Alzheimer’s task are consistent with prior knowledge, as they attempt to do for the AML task. Lastly, the authors should explain the shape of the partial dependence plots (PDPs). This is more of a minor point, but all three PDPs that were shown have the same “checkmark” shape, either upside-down or right-side-up. It is a little concerning to see the same shape so consistently, because it suggests that the DAPr model might not be learning the relationship between meta-features and feature importance as well as suggested. To address this concern, the authors might show the PDPs of different shapes for the other meta-features, explain why the checkmark pattern is correct (and cite it), or explain why the checkmark pattern is an artifact that does not detract from the result.

Correctness: The authors claim that this method avoids the difficulties of traditional feature attribution priors (which require domain knowledge on important features ahead of time), but this claim seems plain wrong. The proposed DAPr method has the same limitation itself, because it requires prior knowledge of which meta-features are important. For example, in both biological tasks, the meta-features (e.g. copy number variations, methylation, etc.) were hand-picked based on strong biological expertise. Since the selection of these meta-features is so important (L206-207), it is inconsistent to claim that the use of metafeatures escapes this domain-knowledge requirement to any reasonable extent. Furthermore, many feature attribution priors rely on very general knowledge, requiring far less domain expertise than the Alzheimer’s/AML examples (e.g. Erion et. al. (2019) suggested using general smoothness of pixels). Additionally, there is an unsupported claim (L242-244) that by looking at the contribution of each meta-feature for a feature’s importance, the DAPr method addresses a limitation of traditional feature attribution methods, where a model produces explanations that are incorrect. The authors have only shown a few examples of features/meta-features that seem to be supported by biological knowledge. There is no global quantification that remotely suggests DAPr reduces the reliance on incorrect features. Also, to claim that the meta-features are likely to be relevant for feature importance is not compelling, because the meta-features are hand-picked specifically to be relevant to the task at hand. In L271-272, the authors claim that there is prior knowledge of a gene’s “hubness” and its relevance for drug response. Is this really known? If so, this needs to be cited. If not, this needs to be removed. Similarly, claims made in L183-190 of the supplement regarding the relationship between gene importance and mutation frequency or methylation are also uncorroborated/uncited.

Clarity: Overall, the flow of the writing is decent, although the myriad of unsupported claims detracts from the reading. Beyond the issues listed above, there are several places which could be clarified, in terms of understandability or providing needed explanations for results. When introducing the concept of meta-features (L40-48), their exact relationship with features could be described more clearly. I did not understand what meta-features are (or how they fit into this method) until later. Showing an example of a meta-feature/feature matrix M (e.g. for movies) might be very helpful. L120 and the equation above have inconsistent notations for g. On these two lines, I think G-hat and ghat should be just g (no hat). On the two-moons experiment, it would be informative to see a visualization of the meta-features for each feature, and a visualization of the global feature importances. I highly suspect that the reason why DAPr works well in this experiment is that the only informative features for all examples is the two coordinate features, and their meta-features are highly distinct from the meta-features dummy noise features. It would be good to confirm this with a 2D scatterplot of the meta-features (i.e. plot mean on one axis and standard deviation on the other, for each of the features). Additionally, it would be good to confirm (as a sanity check) that the network g is actually giving high feature importance to the coordinate features, x1 and x2. The sentence on L260-261 is grammatically incorrect, or at least very difficult to parse. There is no table of performance metrics for the AML drug response task. Since the models with DAPr performed the best, there should be no harm in including it, at least in the supplement.

Relation to Prior Work: The authors have discussed related work on the concept of traditional feature attribution priors, and the use of meta-features for helping predictive models focus on the most important features. However, on L80-81, the sentence summarizing the novel contributions of DAPr is not entirely correct, since methods like Lee et. al. (2018) also use the relationship between meta-features and global feature importance to change how their predictive model is trained. Rather, the main novel contribution of DAPr is that the meta-features are used to change the predictive model’s training specifically as an attribution prior. The other novel contribution made is that the attribution prior is learned from the meta-features using a neural network (as opposed to a linear weighting, as in Lee et. al. (2018)). This should be clarified.

Reproducibility: Yes

Additional Feedback: The idea to learn a feature attribution prior (instead of starting with a known feature attribution prior) is intriguing, and the improved predictive performance speaks for itself. I’ve put my biggest concerns in the “Weaknesses” and “Correctness” sections above, but I will reiterate the following: Please support your claims! There are several grandiose claims that are completely unsupported by analysis or by citations, and others that are dubious if not false. This direly needs to be fixed. Beyond this, I have some additional suggestions/questions. The word “bias” is often used in the text to refer to the DAPr model’s changing how the predictive model is trained. Since the terms “bias” and “biases” can also be nouns, it can be very hard to parse these sentences at first, especially in the abstract. Consider using the verb “drive”, instead. The name “Deep Attribution Priors” could be more descriptive, I think, especially because this term sounds very general, when it refers to a very specific method. The authors might consider an alternative name that refers to the dependence on meta-features, which is a core component of the method (e.g. “Meta-feature-driven Attribution Priors”). Figure 4A is useless. The space should be used for something else. For Figure 4B, several enriched pathways are shown. Are these the top 8 enriched pathways ranked by FDR, or are they picked out among the top pathways?

[Author Response · NeurIPS 2020]

We thank the reviewers for their detailed feedback, and we are encouraged that reviewers found our method (DAPr) to be a "well-motivated" and "novel" approach to solve a "highly relevant" problem to the machine learning community (R2, R4). We are also pleased that R3 found our use of meta-features during model training "clever," and that multiple reviewers appreciated the strength of our empirical results, with improved predictive performance that "speaks for itself" (R4). We will incorporate *all feedback* into the final version of the paper, and address specific reviewer concerns below.

**Additional baselines.** We thank R2 and R3 for their suggestions of additional baselines for comparison! Specifically, as requested by R2, we ran a new experiment on our AML dataset comparing DAPr to RRR [Cite 1]. Since RRR [Cite 1] requires a binary mask designating "relevant" and "irrelevant" features, we considered a gene to be "relevant" if the magnitude of its MERGE score from [Cite 3] was higher than the median magnitude and "irrelevant" otherwise. Additionally, as R3 suggested, we combined DAPr with L1/L2 regularization and we report our results in the table below. *We find that our method, DAPr, still outperforms all of these baselines*, with DAPr + L1 coming close.

| Regularization Method | AML Drug Response |
|---|---|
| Right for the Right Reasons [Cite 1] | $0.889 \pm 0.908$ |
| DAPr + L1 Reg. | $0.804 \pm 0.058$ |
| DAPr + L2 Reg. | $0.859 \pm 0.076$ |
| **DAPr** | $\mathbf{0.786 \pm 0.065}$ |

**Difficulty of collecting meta-features.** Reviewers (R1, R3, R4) were concerned that the applicability of our method might be limited due to the "burdensome" (R3) requirement that a user collect or "craft" (R3) meta-features. In the camera-ready version of our paper, we will clarify that for many problems, such meta-features are *easily obtainable* and *do not need to be handcrafted*. For example, in biomedical problems involving genome-wide data, potential meta-features including gene functions from databases like KEGG or GO, cancer mutations from TCGA, and other epigenomic information such as copy number variation, are easily downloaded from publicly available resources. We note that such meta-features can be used *as-is* with no additional modifications.

**How informative do meta-features need to be?** R3 expressed concern that our method seems to require that all meta-features used are already known to be informative: "the fact that noise priors don't help is not promising when you don't have hand-crafted meta-features." R4 similarly noted that this would limit our method's ability to "escape the domain-knowledge requirement to any reasonable extent." We agree that this would not be "promising," and will better emphasize that *our existing results already show that our framework does not require all meta-features to be informative*. Instead, our framework *learns* to select for informative meta-features from those provided. For example, in our AML results (Fig. 3, main text) we see that hubness and mutation are very informative for feature importance, while other meta-features (e.g. copy number variation, known regulator status) are not. We note that such selections were independently found in [Cite 3] As such, our framework *reduces* the domain knowledge requirement because it can be used with *potentially* relevant meta-features with an *unknown relationship* to feature importance. We thank the reviewers for helping us to better emphasize this, and hope that clarifying that our method aims to *reduce* rather than *remove* the need for domain knowledge will help satisfy R4's concerns about the accuracy of our claims!

**Are attributions forced to be the same across all samples?** R4 expressed concern that "forcing every patient to have the same set of attributions" could mask important signals in the data. We note that our framework does not *force* all samples to have the same attributions, but rather establishes a *prior* on them. The framework is flexible as to how heavily this prior knowledge should be weighted; by tuning the $\lambda$ parameter this prior can be given more or less influence. While, as pointed out by R4, such a prior model may not make sense for some domains (e.g. image classification tasks), previous work has empirically demonstrated benefits in prediction performance from regularizing attributions with a global prior. In a gene-expression context [Cite 2] found benefits from regularizing attributions to values determined by the Laplacian of a gene-gene interaction graph. Moreover, MERGE [Cite 3], the previous state-of-the-art method or incorporating meta-features into model training, uses linear models for prediction which necessarily treat features equally across samples. Given our method's improved performance, we believe our work represents a meaningful step towards incorporating prior knowledge into the training of deep models even with the potential downsides from using a global prior model. We also note that it would be straightforward to extend our framework to incorporate sample-specific information (e.g., sex, age) into our prior models to create even more informative priors.

**Other Concerns.** R4 wrote "there is no table of performance metrics for the AML drug response task"; these metrics can be found in Table 1 in the main text alongside those for the Alzheimer's task. We added citations for the relevance of certain biological features (hubness [Cite 3], methylation [Cite 4]) and will clarify our claims as requested by R4.

**Citations:** [Cite 1] Ross et al., "Right for the right reasons:..." (2017) [Cite 2] Erion et al., "Learning explainable models using attribution priors" (2019) [Cite 3] Lee et al., "A machine learning approach to integrate big data for precision medicine..." (2018) [Cite 4] Moore et al., "DNA Methylation and Its Basic Function" (2013)

[Meta-Review · NeurIPS 2020]

Three referees vote for acceptance, one leans towards rejection. The pros and cons of this paper have been discussed intensively. In my opinion, the rebuttal carefully addressed most of the relevant points of criticism raised in the initial reviews, and my final impression is that this paper does indeed contain relevant contributions, so I also recommend acceptance. However, please consider revising your paper to address reviewer R1’s remark on the prior assumption that every input data point should have the same attributions.